# Predictors of time to death among cervical cancer patients at Tikur Anbesa specialized hospital from 2014 to 2019: A survival analysis

**Benyam Seifu**[1]*, **Chaltu Fikru**[2], **Delelegn Yilma**[3], **Fasil Tessema**[4]

**1** Collage of Medicine and Health Sciences, Ambo University, Ambo, Ethiopia, **2** Faculty of Public Health, Jimma University, Jimma, Ethiopia, **3** College of Medicine and Health Science, Ambo University, Ambo, Ethiopia, **4** Faculty of Public Health, Jimma University, Jimma, Ethiopia

* benyamseifu77@gmail.com

## Abstract

### Background

Cervical cancer (CC) is the 4th most prevalent cancer among females globally. In Ethiopia, around 7,095 new CC cases are diagnosed every year and it is the second common cause of cancer deaths in women. There is limited evidence on survival status as well as about predictors of time to death among CC patients in Ethiopia. Thus, this study investigated the five-year survival status and predictors of time to death among CC patients who had been admitted at Tikur Anbesa specialized Hospital (TASH) from 2014–2019.

### Methods

Facility-based, retrospective-cohort study was conducted at Tikur Anbesa specialized Hospital among 348 patients from June 2014 to June 2019. A systematic random sampling method was employed to select the study participants. Data were extracted from the patient card, and through phone calls. The data was collected using the android version CS-Entry tool. Data was analyzed by SPSS version 23. Kaplan and Meier's method was used to estimate survival functions and Cox-proportional hazard regression analysis was carried out in order to identify the independent predictors of time to death.

### Results

The overall incidence of death was 31 per 100 person-years of follow up. The median (IQR) follow-up time of the entire cohorts was 18.55 (8.96–49.65) months. The independent predictors for time to death included; age older than 50 years [AHR: 1.4; 95% CI: 1.1–1.9], late stage of CC at diagnosis [AHR: 2.2; 95% CI: 1.7–2.9], No CC treatment [AHR: 2.1; 95% CI: 1.5–3.1] and HIV positive [AHR: 2.3; 95% CI: 1.4–3.8].

### Conclusion and recommendation

The death rate of CC patients was high. The significant predictors associated with shorten time to death of CC patients were older age, advanced cancer stage at diagnosis, HIV

**Data Availability Statement:** The data relevant to this study cannot be shared as it contains potentially identifying and sensitive clinical, medical, and pathological information. Interested

researchers can send requests for data to Jimma University research ethics review committee. Requests can be directed to Gudina Terefe Tucho via e-mail address: gudina.terefe@ju.edu.et. The collected data for this study are also available from the principal author and will be available upon request.

**Funding:** The authors received no specific funding for this work.

**Competing interests:** The authors have declared that no competing interests exist.

infection and not receiving cancer treatment. Therefore, improving early detection and initiation of treatment for all CC patients is necessary in order to improve patient's survival status. The government needs to strengthen the routine CC screening programs to address high-risk women such as elderly and HIV positive women in Ethiopia.

## Introduction

Cervical cancer (CC) is one of the gravest threats to women's lives. It is a malignant neoplasm of the cervix uteri. It may be completely asymptomatic in early stages [1]. However, it may present as persistent pelvic pain, unexplained weight loss, bleeding between periods and after sexual intercourse, unusual vaginal discharge, and pain after sexual intercourse in its advanced stage [2]. The primary cause of cervical pre-cancer and cancer is persistent or chronic infection with one or more of the "high-risk" (or oncogenic) types of human papillomavirus (HPV) [1,3,4]. Studies have estimated that over 80% of sexually active women could be infected with genital HPV at some point in their lifetime [1].

Cervical cancer occurs worldwide and 1 out of 8500 women are affected by CC worldwide annually [2], An approximate estimation of CC in 2018 indicated that, 570 000 cases of cervical cancer and 311 000 deaths from CC occurred globally [5]. However, the highest incidence rates are found in Central and South America, East Africa, South and South-East Asia, and the Western Pacific. Over the past three decades, the burden of cancer has shifted to less developed countries, as this developing countries share 57% of global CC cases and 65% of global CC deaths [6].

Even though CC is preventable, its incidence and mortality is still high in developing countries [7]. It is the leading cause of cancer-related death in women in eastern, western, middle, and southern Africa [5]. Cervical cancer treatment in Africa is hampered by the lack of diagnostic and treatment facilities, lack of healthcare infrastructure and poor pathology services. Further, there is a significant brain drain of trained healthcare workers in Africa that exacerbates the problem [7]. It is estimated that by the year 2030, CC will kill more than 443,000 women yearly worldwide, most of them in sub-Saharan Africa [8]. According to WHO, many of those who died of CC are breadwinners, and care takers for both the children and elders. In sub-Saharan African countries, women are the head of one-third of all households and over half of the children who have lost a parent are being cared by grandmother are also at risk of cervical cancer. In addition to emotional trauma, CC deaths have significant economic costs over the short and long-term. Family members may lose work opportunities and can incur overwhelming medical costs while caring for women with cancer [9].

In Ethiopia, cancer accounted for about 5.8% of total national mortality. In Addis Ababa, the annual incidence of cancer is estimated to be around 60,960 cases and the number of annual deaths reached over 44,000. Data from the radiotherapy center at Tikur Anbesa Specialized Hospital (TASH) showed that, CC ranks as the second leading cause of female cancer among patients visiting the oncology center [4]. Based on the World Health Organization (WHO)/Institut Català d'Oncologia (ICO) Information Centre report, Ethiopia has a population of 20.90 million women ages 15 years and older who are at risk of developing CC. Farther more, one out of four women started sexual intercourse before age 15 [10]. This believed to increase the risk of getting HPV, the causative agent of CC [11]. Moreover, studies have shown that the level of CC screening in Ethiopia was less than 10% [11–13].

The above statistics about Ethiopia shows that, many women are at risk of CC. Even though some researches are conducted on CC, most of the researches conducted in Ethiopia focused on the knowledge and practice of cervical cancer screening and are from 2008–2012 [12,14]. Those studies also didn't give emphasis to the stage of cancer at which the patient presents, which is one of the major determinants for the prognosis of CC. Besides stage of CC, several factors such as educational status, financial capability, location and presence of health care facilities determine the stage at which patients with cancer present to the health facility needs further investigation [15]. Furthermore, improved health services, cervical cancer screening campaigns, health professional's competency and other related changes are expected to affect the survival of cervical cancer patients [1]. For those all reasons, it is very significant to study the current survival status of CC patients and predictors of time to death among CC patients. Therefore, this study investigated the current survival status and predictors of time to death among CC patients in Ethiopia.

## Methods

### Study setting and study population

The study was conducted in the Tikur Anbesa Specialized Hospital in Addis Ababa, the capital city of Ethiopia. It is the largest referral hospital located in the center of the city. The hospital has a total of 600 beds; of which 18 beds are allocated to adult cancer patients' and 26 beds are used to pediatric oncology and hematology. The study was conducted using facility-based retrospective-cohort study design. The study populations were all women who had been diagnosed with CC and enrolled from June 2013- until June 2014. Patients who were admitted to TASH during the inclusion period, but their first date of diagnosis with CC was before the inclusion period, patients or collateral phone number was not recorded and incomplete patient charts on the day of diagnosis and stage of CC were excluded from the study.

### Sample size determination and sampling procedure

The sample size was calculated based on the assumption that the type I error 5%, power of 90%, the median survival time among exposed (late stage CCP) was 30 months and the median survival time among unexposed (early stage CCP) was 50 months, median survival times were taken from previous study conducted in black lion [14]. If we assumed that patients enter into the study at a uniform rate over a T- time period, in this study 60 months (5 years), Gross and Clark developed the following formula which depends on the separate median survival times for the exposure and non-exposure [16]. The required patients in each group were calculated as follows.

$$n = (Z_\alpha + Z_\beta)^2 [\Phi(\mu_E) + \Phi(\mu_C)]/(\mu_E^{-1} - \mu_C^{-1})^2$$

where

$$\Phi(\mu_i) = \frac{T}{\mu_i^3} / \left[\frac{T}{\mu_i} - 1 + \exp(-T/\mu_i)\right], i = C, E$$

C = median survival time for non-exposed group (50 months), E = median survival time for exposed group (30 months) [14], T = total time in which study subjects are recruited/ entered to the study (60 months), $\alpha$ = level of significance (0.05), Z$\alpha$/2 = 1.96 at 95% confidence interval, Power = 1-$\beta$ = 90%, Z$\beta$ = 1.28 and n = minimum sample size for each group. The minimum sample size required for each group was 158. After adding 10% for incomplete records the final sample size was 174 for exposed (Late stage CCP) and 174 for non-exposed (early stage CCP) and the total sample size was 348.

The systematic random sampling procedure was used to select patient cards. From the data obtained from TASH registry, a total of 710 patients were diagnosed with CC from June 2013-until June 2014. The interval K-Value was calculated by dividing 710/348 = 2.04. So we reviewed every two other CCP card until the expected sample size was obtained. The study participants were retrospectively followed from the date of diagnosis for five years. The starting point for the retrospective follow-up was the date of diagnosis with CC and the endpoint of the follow up was the date of death (from patient card or by phone call), date of lost to follow up (last visit or of last contact) or the end date of follow-up period (June, 2019).

## Data collection tool and procedure

Record review tool was prepared by reviewing different literatures and patient charts. The review tool has socio-demographic, past obstetric and medical history sections. Before collecting the data, the records were examined whether they full fill the eligibility criteria or not. The data were collected by reviewing records from patients' registration book and individual follow-up chart using pretested data collection form. The data were collected electronically using android tablet. The questionnaire template was coded by using open source software for Computer Assisted Personal Interviewing using census and survey processing system (CS-Pro) version 7.1 and deployed to Census and survey entry (CS-Entry) android application to collect the data from patient card. Data were collected by two registered nurses who are not working at TASH. One senior oncology nurse supervised the day to day data collection and cross checked consistency of collected data. Principal investigator controlled overall data collection and other research process.

## Data analysis

The data collected through CS-Entry android app was exported to SPSS version 23 for data cleaning and analysis. Basic descriptive analyses were done and presented as frequency and percent for categorical variable. Continuous variables were reported with mean and standard deviation (SD) and in terms of median (inter quartile range). Kaplan Meier survival curve with a log-rank test was fitted in order to evaluate the presence of a difference in survival time among various predictor variables. The correlation between survival time and the covariates was analyzed using cox regression model. Kaplan Meier analysis method was used to estimate survival functions and Cox-proportional hazard regression analysis was carried out to identify independent predictors of time to death. Multi-collinearity was checked by using variance inflation factor. The proportionality of hazard assumption was checked using the Log (-Log) S (t) plots. The crude and adjusted hazard ratios with their 95% Confidence Intervals (CI) were estimated, and p-values less than 0.05 were used to declare the presence of statistically significant correlation between predictors and survival time. Finally, the results were presented using text, tables and figures.

## Operational definitions

Survival of CC patients is defined as survival time from the first confirmed diagnosis date of cervical cancer, to death [14]. CC patients are women who are diagnosed as cervical cancer, which doesn't include pre-cancers lesions. Early stage CCP (ESCCP) is defined as stage I and II and Late stage CCP (LSCCP) is defined as Stage III and IV [1].

### Ethical statement

Ethical clearance was received from the Institutional Review Board committee (IRB) of Jimma University (reference number: IHRPG/644/2019). Research proposal was also submitted to Tikur Anbesa specialized hospital oncology department and approval letter was obtained to conduct the study in the hospital. Due to the unavailability of the CC patients at TASH during the data collection, informed oral consent was taken. Both of the institutions approved the use informed oral consent which was taken from study participants through phone call. For those study participants who died, we used the care giver phone number which was registered in the patient card and they have given oral consent. For those who agreed to participated in the study, patients cards were retrospectively reviewed form June,01, 2013 to June, 01, 2019 to collect information regarding patients socio-demographic, past obstetric history, medical history and treatments they have received.

## Results

### Socio-demographic characteristics

From the 348 reviews to be done only two patients refuse to participate during phone call and gave a total response rate of 99.4%. The mean ± standard deviation (SD) of participant's age was 50 ± 11 years. Majority of the study participants, 214 (73.8%) were married and 239 (69.1%) were rural residents. Regarding educational status, 128 (37.0%) of the study participants didn't attend any formal education. Half of the study participants 175 (50.1%) were housewives with respect to the occupational status (Table 1).

### Past obstetrics and medical history

Concerning the past obstetrics history, 197 (56.9%) of study participants had five and more children. Most of the study participants 324 (93.6%) they didn't have any non-communicable disease. Among the total of 141(40.7%) cervical cancer patients who tested for HIV, 30 (21.2%) were found to be HIV positive and 111 (78.8%) were found to be HIV negative. From those, 24 (10.8%) HIV positive and 61 (27.4%) HIV negative patients were died during the follow up period (Table 2).

### Clinical and pathological characteristics

From the total of 223 (64.5%) cervical cancer patients died during the follow up period, 95 (42.6%) were early stage and the rest 128 (57.4%) were late stage. Majority of the study participants, 296 (86.8%) had a well differentiated histological grade and 326 (94.2%) had Squamous cell carcinoma. Regarding the treatment, 303 (87.6) had started treatment and 20 (5.8%) had surgery, 103 (29.8) had Chemotherapy with radiotherapy and 130 (37.6) had radiotherapy alone (Table 3).

### Incidence of death during the follow-up

The median (IQR) follow-up time was 18.55 (8.96–49.65) months; [28.6 (12.62–51.57) for early stage and 11.6 (7.25–17.82) for late stage cervical cancer at diagnosis] with total follow-up time of 738.06 years (467.13 for early stage and 270.93 for late stage). The overall death rate was 31 per 100 person-years of follow up; 21 per 100 person-years of follow up among early stage and 48 per 100 person-years of follow up among late stage, respectively.

**Table 1. Socio-demographic characteristics of CCP in TASH from July 2014 –July 2019.**

| Characteristics | Censored (n = 123) Number (%) | Dead (n = 223) Number (%) | Total (N = 346) Number (%) |
|---|---|---|---|
| Age in years | | | |
| ≤39 | 24 (19.5) | 26 (11.7) | 50 (14.5) |
| 40–49 | 51 (41.5) | 60 (26.9) | 111 (32.1) |
| 50–59 | 30 (24.4) | 73 (32.7) | 103 (29.7) |
| ≥ 60 | 18 (14.6) | 64 (28.7) | 82 (23.7) |
| Marital status | | | |
| Married | 71 (77.2) | 143 (72.2) | 214 (61.8) |
| Widowed | 16 (17.4) | 39 (19.7) | 55 (15.9) |
| Other* | 5 (5.4) | 16 (7.8) | 21 (6.1) |
| Unknown | 31 (25.2) | 25 (11.2) | 56 (16.2) |
| Educational status | | | |
| No formal education | 42 (34.1) | 86 (38.5) | 128 (37.0) |
| Primary education | 5 (4.1) | 19 (8.5) | 24 (7.0) |
| Secondary and above | 5 (4.1) | 10 (4.5) | 15 (4.3) |
| Unknown | 71 (57.7) | 108 (48.4) | 179 (51.7) |
| Occupation | | | |
| Housewife | 60 (48.8) | 115 (51.6) | 175 (50.1) |
| Farmer | 13 (10.6) | 29 (13.0) | 42 (12.1) |
| Private employee | 41 (33.3) | 60 (26.9) | 101 (29.2) |
| Other** | 9 (7.3) | 19 (8.5) | 28 (8.1) |
| Place of residence | | | |
| Urban | 38 (30.9) | 69 (30.9) | 107 (30.9) |
| Rural | 85 (69.1) | 154 (69.1) | 239 (69.1) |

* Single, divorced

** Government employee, Merchant.

## Survival time among different groups

The overall median survival time of the study participants from the Kaplan and Meier survival analysis was 17.6 months (95% CI: 14.0–19.2). The median survival time between stages of cancer showed a significant difference with 28.6 months (95% CI: 23.7–33.4) among early stage and 1.6 months (95% CI: 10.4–12.5) among late stage patients. Regarding age, the median survival time of study participants with age 50 and younger was significantly higher than those with age older than 50 years; 19.2 months (95% CI: 13.5–18.2) and 15.8 months (95% CI: 14.1–19.3) respectively. Study participants who received any cancer treatment during the follow up period had a median survival time of 17.9 months (95% CI: 13.33–22.4) while the median survival time for study participants who didn't receive any treatment was 9.1 months (95% CI: 6.9–11.3). The median survival time of study participants who had surgery was found to be significantly higher than those who don't have surgery, 42.4 months (95% CI: 34.9–49.9) and 27.6 months (95% CI: 25.1–30.1) respectively. The median survival time of study participants whose HIV status found to be positive was significantly lower than those with HIV negative; 9.7 months (95% CI: 7.4–12.1) and 24.9 months (95% CI: 14.5–35.3) respectively (Table 4).

Cancer patients who were diagnosed at early stages (stage I and II) lived for longer time than patients diagnosed at late stages (stage III and IV), P-Value = 0.0001 (Fig 1). Cancer patients who received any cancer treatment (surgery/chemotherapy/radiotherapy) survived for longer duration than those patients who did not received any of the cancer treatment

**Table 2. Past obstetrics and medical history of CCP in TASH from July 2014 –July 2019.**

| Characteristics | Censored (n = 123) Number (%) | Dead (n = 223) Number (%) | Total (N = 346) Number (%) |
|---|---|---|---|
| Number of pregnancy | | | |
| < 5 | 52 (42.3) | 97 (43.5) | 149 (43.1) |
| 5 ≥ | 71 (57.7) | 126 (56.5) | 197 (56.9) |
| History of abortion | | | |
| Yes | 6 (4.8) | 7 (3.1) | 13 (3.8) |
| No | 64 (52.1) | 102 (45.7) | 166 (48.0) |
| Unknown | 53 (43.1) | 114 (51.1) | 167 (48.2) |
| Age of first sexual intercourse | | | |
| < 18 | 7 (5.7) | 33 (14.8) | 40 (11.6) |
| ≥18 | 4 (3.3) | 5 (2.2) | 9 (2.6) |
| Unknown | 112 (91.1) | 185 (83) | 297 (85.8) |
| Number of sexual partners | | | |
| One | 83 (67.5) | 173 (77.6) | 256 (74.0) |
| Multiple | 8 (6.5) | 20 (9) | 28 (8.1) |
| Unknown | 32 (26.0) | 30 (13.5) | 62 (17.9) |
| HIV status | | | |
| Positive | 6 (4.9) | 24 (10.8) | 30 (8.7) |
| Negative | 50 (40.7) | 61 (27.4) | 111 (32.1) |
| Unknown | 67 (54.5) | 138 (61.9) | 205 (59.2) |
| Presence of Co-morbid disease | | | |
| Yes | 7 (5.7) | 15 (6.7) | 22 (6.4) |
| No | 116 (94.3) | 208 (93.3) | 324 (93.6) |

options, P-Value = 0.0001 (Fig 2). Cancer-HIV comorbid patients had shorter survival period compared to their counter parts, P-Value = 0.002 (Fig 3).

## Predictors of time to death among cervical cancer patients

In order to identify predictors of time to death, Cox proportional regression model was used. Before fitting the covariate into the model, proportional hazard assumption was checked by examining Log (-Log S (t)) plots. Overall seven variables; age, stage of cancer, received treatment, chemotherapy, radiotherapy, surgery and HIV status were found to be independently and significantly associated with time to death.

In the final Cox proportional regression model, four variables were found to be significantly associated with time to death with. Patients who were older than 50 years were 1.4 times more likely to die within five years [AHR: 1.4 (95% CI: 1.1–1.9)]. Patients diagnosed at late stage were 2.2 (95% CI: 1.7–2.9) time more likely to die earlier than those diagnosed at early stage. Likewise, those who didn't receive any kind of treatment were two times more likely to die within five years [AHR: 2.1 (95% CI: 1.5–3.1)]. In addition, those who were HIV positive were two times more likely to die than HIV negative patients [AHR: 2.3 (95% CI: 1.4–3.8)] (Table 5).

## Discussion

This study investigated the five-year survival status and predictors of time to death among CC patients in Ethiopia. In this study, the median survival time of CC patients was 17.6 months. There were significant differences in the median survival time between categories of covariates like age, stages of cervical cancer, received treatment, surgery and HIV Status. The death rate

**Table 3. Clinical and pathological characteristics among CCP in TASH from July 2014 –July 2019.**

| Characteristics | Censored (n = 123) Number (%) | Dead (n = 223) Number (%) | Total (N = 346) Number (%) |
|---|---|---|---|
| Stage of cancer at diagnosis | | | |
| Early Stage | 79 (64.2%) | 95 (42.6%) | 174 (50.3) |
| Late Stage | 44 (35.8%) | 128 (57.4%) | 172(49.7) |
| Histological grade | | | |
| Well differentiated | 106 (86.2%) | 190 (87.2%) | 296 (86.8) |
| Moderately differentiated | 11 (8.9%) | 18 (8.3%) | 29 (8.5) |
| Poorly differentiated | 6 (4.9%) | 10 (4.6%) | 12 (3.5) |
| Histological Type | | | |
| Squamous cell carcinoma | 114 (94.3%) | 212 (95.1%) | 326 (94.2) |
| Adenocarcinoma | 9 (7.3%) | 11 (4.9%) | 20 (87.8) |
| Received any cancer treatment | | | |
| Yes | 116 (94.3%) | 187 (83.9%) | 303 (87.6) |
| No | 7 (5.7%) | 36 (16.1%) | 43 (12.4) |
| Surgery | | | |
| Surgery | 14 (11.4%) | 6 (2.7%) | 20 (5.8) |
| No surgery | 109 (88.6%) | 217 (97.3%) | 326 (94.2) |
| Chemotherapy | | | |
| Chemotherapy alone | 7 (5.7%) | 13 (5.8%) | 20 (5.8) |
| Chemotherapy & radiotherapy | 37 (30.1%) | 66 (29.6%) | 103 (29.8) |
| Chemotherapy & other therapy | 5 (4.1%) | 2 (0.9%) | 7 (2.0) |
| No chemotherapy | 74 (60.2%) | 142 (63.7%) | 216 (62.4) |
| Radiotherapy | | | |
| Radiotherapy alone | 40 (32.5%) | 90 (40.4%) | 130 (37.6) |
| Radiotherapy & palliative care | 6 (4.9%) | 12 (5.4%) | 18 (5.2) |
| No radiotherapy | 77 (62.6%) | 121 (54.3%) | 198 (57.2) |

of CC patients was 31 per 100 person-years follow up. This finding is higher than CC mortality incidence rate reported in Sub- Saharan Africa (17.9/100/year) [7]. This difference could be due to variations in the study period, the cancer stage at presentation, in waiting time for treatment after diagnosis and difference in quality of cancer care services [12,17]. Treatment of cervical cancer in Ethiopia is hampered by the lack of diagnostic and treatment facilities, lack of healthcare infrastructure and poor pathology services. This results in long waiting times and cause many potentially curable tumors to progress to incurable stages and premature death [18,19]. Literatures also indicated that CC patients in developing countries have additional comorbidities and lower survival than those in higher income countries [20]. The independent predictors significantly associated with shorten time to death of cervical cancer patients were older age, advanced cancer stage at diagnosis, HIV infection and not receiving cancer treatment.

This study hypothesized that stage of cancer at time of diagnosis is the key predictor for the survival of CCP. Advanced stage of cancer is significantly associated with lower survival of cervical cancer patients [1,2]. This study also showed that LSCCP are two times more likely to die and their median survival time is significantly shorter than ESCCP [AHR: 2.2 (95% CI: 1.7–2.9)]. The study conducted in TASH also revealed that, CCPs who are stage-IV are three times more at risk of dying when they are compared with stage I CCPs [14]. This finding also supported by studies conducted in Germany, Estonia and Australia, reported that cancer stage is a significant prognostic factor for survival of CCPs [21–23]. The higher mortality in patients

**Table 4. Survival time among different groups of CCP in TASH, June 2014 to June 2019.**

| Characteristics | Dead (n = 223) | Median survival time Estimate (95% CI) | log rank $X^2$-value | P-value |
|---|---|---|---|---|
| Age | | | | |
| ≤50 years | 86 (38.6%) | 19.2 (13.5–18.2) | 7.9 | 0.005 |
| >50 years | 137 (61.4%) | 15.8 (14.1–19.3) | | |
| Residence | | | | |
| Urban | 69 (30.9%) | 18.6 (10.8–26.4) | 0.4 | 0.555 |
| Rural | 154 (69.1%) | 16.1 (13.7–18.6) | | |
| Stage at diagnosis | | | | |
| Early | 95 (42.6%) | 28.6 (23.7–33.4) | 34.3 | 0.0001 |
| Late | 128 (57.4%) | 11.6 (10.4–12.5) | | |
| Received treatment | | | | |
| Yes | 187 (83.9%) | 17.9 (13.3–22.4) | 18.5 | 0.0001 |
| No | 36 (16.1%) | 9.1 (6.9–11.3) | | |
| Chemotherapy | | | | |
| Yes | 81 (36.3%) | 17.8 (122–23.4) | 0.6 | 0.443 |
| No | 142 (63.7%) | 15.5 (122–18.8) | | |
| Radiotherapy | | | | |
| Yes | 102 (45.7%) | 14.1 (11.0–17.1) | 2.3 | 0.127 |
| No | 121 (57.3%) | 17.8 (12.8–22.7) | | |
| Surgery | | | | |
| Yes | 6 (2.7%) | 27.6 (25.1–30.1) | 9.3 | 0.002 |
| No | 217 (97.3%) | 15.8 (13.7–17.9) | | |
| HIV Status | | | | |
| Positive | 24 (10.3%) | 9.7 (7.4–12.1) | 12.9 | 0.002 |
| Negative | 61 (27.4%) | 24.9 (14.5–35.3) | | |
| Unknown | 139 (62.3%) | 13.9 (11.2–16.5) | | |
| Total | 223 (100.0%) | 17.6 (14.0–19.2) | | |

with advanced cancer stage could be attributed to the rapid metastasis rate, increase the risk of developing comorbid diseases and treatment complications [1,2]. Furthermore, this could be due to fact that patients with advanced stage are less likely to respond to treatment than their early stage counterparts [24].

In this study, advanced age was found to be one of the predictor of CCP survival [AHR: 1.4; 95% CI: 1.1–1.9]. Women who are older than 50 years are two times more likely to die within five years than women who are 50 and younger at diagnosis. Previous studies also reported that older age at diagnosis is associated with lower survival time of cervical cancer patients [14,17,25]. A study from japan revealed that, advanced stage at diagnosis was the main determinant of poor survival among the aged CC patients [25]. Several studies also found poor cervical cancer prognosis and higher rate of mortality among older women with cervical cancer [26–28]. This may be due to the presence of more advanced disease among older women at diagnosis [29], and older women receive less aggressive treatment as compared to their younger counterparts [27]. But in contrast, study from china reported that CC has the same prognosis in old and young women [30]. This difference may be due to the design of the study and difference in age classification.

It's scientifically known that treatments prolong patient's survival time. This study also evidenced that patients who didn't received treatment died twice more than who received treatment. Different literatures also support this finding [12,31–33]. Treatment includes surgery,

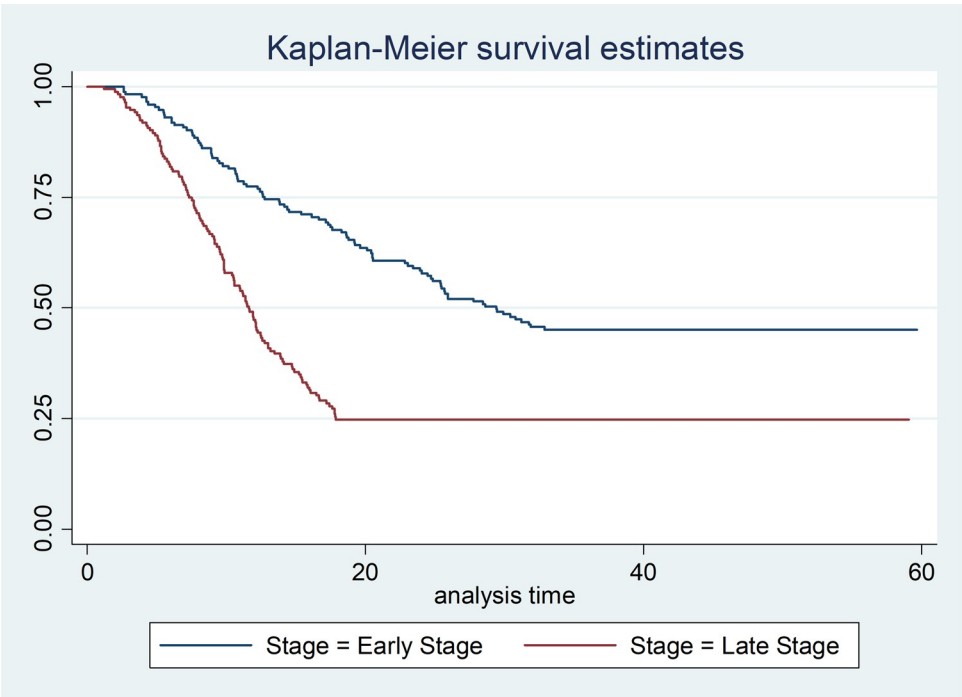

**Fig 1.**

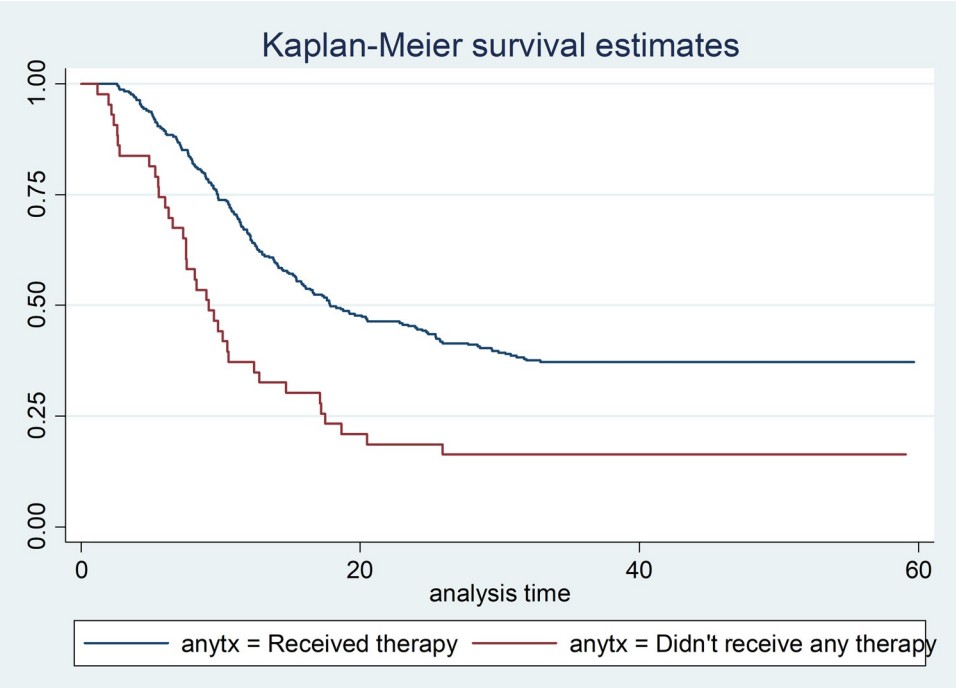

**Fig 2.**

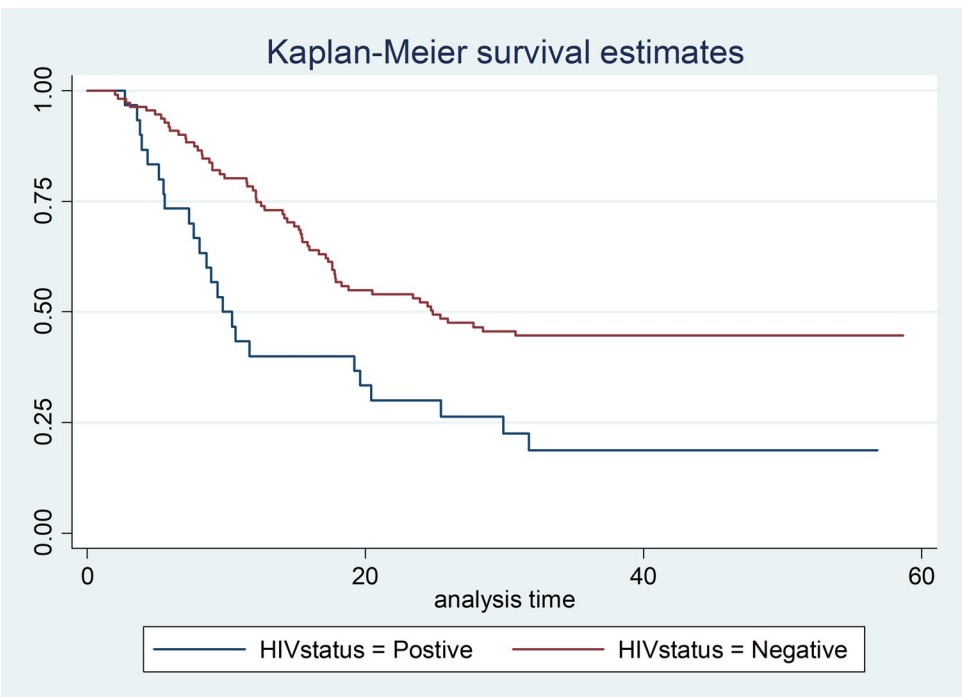

**Fig 3.**

Chemotherapy, Radiotherapy, and or combination of the above [1]. Unlike other similar studies, our study didn't show which specific treatment increases the survival of CCP. However, during KM analysis we found a significant difference in median time with CCP who had surgery and those who didn't have surgery. This was also found in other similar study conducted in TASH five years ago [14]. One of the reason that surgery didn't come as a predictor during multiple Cox-proportional hazard model might be due to cofounded with variables like stage of cancer and received any treatment.

In this study, from the study participants whose HIV status was written on their patient card, HIV positives tend to die twice more likely than HIV negatives. The possible scientific explanation is that HIV decreased efficacy of cellular immune responses to preserve oncologic remission, decreased tolerance of the chemotherapy as well as the radiation therapy, and increased prevalence of anemia in HIV infected women, which impairs efficacy of radiation [34,35]. This is evidenced by different studies like the study from Brazil showed, CCP who were treated and had a complete response, after two years of diagnosis of CC HIV was found to increase the incidence of mortality (HR: 2.2, 95% CI: 1.3–3.2, but within the first two years of diagnosis, there was no significant difference in survival of HIV positives and HIV negative [36]. The study from Botswana also revealed that HIV infection significantly increased the risk for death among all women (AHR, 1.9; 95% CI, 1.2 to 3.2) [37]. This finding was also supported by study conducted in Kenya [35].

## Limitations of the study

The study was conducted in the largest chemotherapy and radiotherapy center in Ethiopia, and which more advanced and complicated patients are referred to the oncology center of TASH. However, the incidence of death may be undermined by absence of survival data on the patient card, patient or collateral phone number is unavailable in the card, hence we,

**Table 5. Cox regression analysis of predictors of time to death among CCP in TASH, 2014–2019.**

| Characteristics | Status | | CHR (95% CI) | AHR (95% CI) |
|---|---|---|---|---|
| | Censored | Dead | | |
| Age | | | | |
| ≤50 years | 75 (46.6%) | 86 (53.4%) | 1 | 1 |
| >50 years | 48 (25.9%) | 137 (74.1%) | 1.5 (1.1–1.9) ** | 1.4 (1.1–1.9) * |
| Stage of cancer | | | | |
| Early | 79 (45.4%) | 95 (54.6%) | 1 | 1 |
| Late | 44 (25.6%) | 128 (74.4%) | 2.2 (1.7–2.9) *** | 2.2 (1.7–2.9) *** |
| Received treatment | | | | |
| Yes | 116 (38.3%) | 187 (61.7%) | 1 | 1 |
| No | 7 (16.3%) | 36 (83.7%) | 2.2 (1.5–3.1) *** | 2.1 (1.5–3.1) *** |
| Chemotherapy | | | | |
| Yes | 49 (37.7%) | 81 (62.3%) | 1 | |
| No | 74 (34.3%) | 142 (65.7%) | 1.1 (0.8–1.5) | |
| Radiotherapy | | | | |
| Yes | 46 (31.1%) | 102 (68.9%) | 1 | |
| No | 77 (38.9%) | 121 (61.1%) | 1.2 (0.9–1.6) | |
| Surgery | | | | |
| Yes | 14 (70.0%) | 6 (30.0%) | 1 | 1 |
| No | 108 (33.4%) | 217 (66.6%) | 3.3(1.5–7.4) ** | 1.9 (0.9–4.6) |
| HIV Status | | | | |
| Negative | 50 (45.0%) | 61 (55.0%) | 1 | 1 |
| Positive | 6 (20.0%) | 24 (80.0%) | 2.2 (1.4–3.5) ** | 2.3 (1.4–3.8) ** |
| Unknown | 67 (32.7%) | 139 (67.3%) | 1.5 (1.1–2.1) ** | 1.5 (1.1–2.1) |

NB

* p-value<0.05

** p-value<0.01

***P-value< 0.001.

considered them as lost follow up. Incomplete information concerning important variables like CC screening, nutritional status, age at starting sexual intercourse, number of sexual partner and substance use were not found, which might co-found the result. The exact day of lost follow up is unknown that may reduce the median survival time. Since the data is based on secondary data, the reliability of the data relay on the data on the cancer patient card. And unable to get adequate information about cause of death to identify the actual cause of death for those patients reported as dead.

## Conclusion and recommendation

The death rate of CC patients was found to be high (31 per 100 person-years of follow up). There were significant differences in the median survival time between categories of covariates like age, stages of cervical cancer, received treatment, surgery and HIV Status. The significant predictors associated with time to death of cervical cancer patients were age, stage of cancer, receiving treatment and HIV status. We recommend health care providers to initiate early treatment for all CC patients in order to improve their survival status and strengthen the routine CC screening programs for high-risk women such as elderly and HIV positive women. We also recommend researchers to conduct a prospective study in order to appropriately estimate the survival time of cervical cancer patients. It is also necessary to strengthen the already

started HPV vaccination and early screening of cervical cancer among all women at risk in Ethiopia.

## Acknowledgments

### Declarations

We would like to thank Jimma University for facilitating to conduct this study. We also thank Tikur Anbesa specialized hospital oncology staffs for their cooperation during data collection process. We will like to memorize Dr Argaw Berhei who passed away recently by the current pandemic. He was TASH oncology department head and he played a valuable assistance on reviewing the proposal and approving the during data collection.

## Author Contributions

**Conceptualization:** Benyam Seifu, Chaltu Fikru, Delelegn Yilma, Fasil Tessema.

**Data curation:** Benyam Seifu, Chaltu Fikru, Fasil Tessema.

**Formal analysis:** Benyam Seifu, Chaltu Fikru, Delelegn Yilma.

**Funding acquisition:** Benyam Seifu.

**Investigation:** Benyam Seifu, Chaltu Fikru.

**Methodology:** Benyam Seifu, Chaltu Fikru, Delelegn Yilma, Fasil Tessema.

**Project administration:** Benyam Seifu.

**Resources:** Benyam Seifu.

**Software:** Benyam Seifu, Chaltu Fikru, Fasil Tessema.

**Supervision:** Benyam Seifu, Chaltu Fikru, Delelegn Yilma, Fasil Tessema.

**Validation:** Benyam Seifu, Chaltu Fikru.

**Visualization:** Benyam Seifu, Fasil Tessema.

**Writing – original draft:** Benyam Seifu, Delelegn Yilma, Fasil Tessema.

**Writing – review & editing:** Benyam Seifu, Chaltu Fikru, Delelegn Yilma, Fasil Tessema.

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
