## [Decision Letter · Decision Letter 0]

11 Oct 2021

PONE-D-21-03938Predictors of time to death among Cervical Cancer patients at Tikur Anbesa Specialized Hospital from 2014 to 2019: A Survival AnalysisPLOS ONE

Dear Dr. Seifu,

Thank you for submitting your manuscript to PLOS ONE. After careful consideration, we feel that it has merit but does not fully meet PLOS ONE’s publication criteria as it currently stands. Therefore, we invite you to submit a revised version of the manuscript that addresses the points raised during the review process.

Please consider the external reviewer's comments carefully and use these comments to revise you manuscript.  These comments are considered "minor" and your manuscript will be recommended for acceptance after you revise your manuscript.

We look forward to receiving your revised manuscript.

Kind regards,

James P Brody

Academic Editor

PLOS ONE

Journal Requirements:

3. In the Methods, please clarify that participants provided oral consent. Please also state in the Methods:

- Why written consent could not be obtained

- Whether the Institutional Review Board (IRB) approved use of oral consent

- How oral consent was documented

- How consent was managed from patients who had died; please state whether the ethics committee waived consent in some cases.

For more information, please see our guidelines for human subjects research: https://journals.plos.org/plosone/s/submission-guidelines#loc-human-subjects-research

4. In the ethics statement in the manuscript and in the online submission form, please provide additional information about the patient records/samples used in your retrospective study, including the date range (month and year) during which patients' medical records/samples were accessed.

5. Thank you for stating the following financial disclosure: "This study not funded by any institutions/organizations" 

7. Your ethics statement should only appear in the Methods section of your manuscript. If your ethics statement is written in any section besides the Methods, please move it to the Methods section and delete it from any other section. Please ensure that your ethics statement is included in your manuscript, as the ethics statement entered into the online submission form will not be published alongside your manuscript. 

Reviewers' comments:

Reviewer's Responses to Questions

**Comments to the Author**

1. Is the manuscript technically sound, and do the data support the conclusions?

Reviewer #1: Partly

2. Has the statistical analysis been performed appropriately and rigorously? 

Reviewer #1: Yes

3. Have the authors made all data underlying the findings in their manuscript fully available?

Reviewer #1: Yes

4. Is the manuscript presented in an intelligible fashion and written in standard English?

Reviewer #1: No

5. Review Comments to the Author

Reviewer #1: General comments

The study “Predictors of time to death among Cervical Cancer patients at Tikur Anbesa Specialized Hospital from 2014 to 2019: A Survival Analysis” by Seifu and colleagues is interesting and to the literature of cervical cancer in Ethiopia and Africa. However there are few things to consider to improve this manuscript. First the wording and grammar should be improved as it is very difficult to read. I suggest a native speaker to edit this manuscript. Also the introduction is not structured very well and needs to be reworked. The methods could be improved and the discussion should be structured. Below are my comments which should be addressed.

Abstract

Introduction:

Authors should provide the full meaning of any abbreviations at the first use. Authors should write out the full meaning of TASH at first use. The phrase in the objective that indicate that authors assess is wrong. Authors investigated or determined. Access is not the right word to use.

Methods:

The statement “..sample of 348 patients under follow-up time…” should be rephrased. Authors followed 348 women or a cohort of 348 women. Follow-up time seems vague and not standard. And it is imperative for authors use the phrase “348 patients” rather than a sample of 348 patients. This should be corrected throughout the whole manuscript.

The sentence “The data was collected and entered using the android version CS-Entry tool and for the analysis exported to SPSS version 23” should be broken into two and made clear. Just say data was analyzed by SPSS version 23. Authors should keep their sentences simple and clear.

Conclusions:

The sentences “As the stage of cancer progressed, the chance of surviving gets reduced. Therefore, due emphasis should be given on improving early detection. Routine CC screening programs for high-risk women such as elderly and HIV positive women should be strengthened” should be re written and made more clear and concise. Phrases like “surviving gets reduced” is not scientific and a should be written in a better way.

Introduction

Line 65- authors should put “early” in front of screening

The first paragraph of the introduction is succinctly written and the epidemiology is clearly stated. Therefore, I suggest authors delete their second paragraph as it only elaborately gives the epidemiology of Cervical cancer and adds no new information. It makes the introduction unnecessary long and adds no value besides the manuscript is not focused on the epidemiology of CC.

Lines 81-85 already stated in the first paragraph and must not be repeated.

Line 89: what is grandmother women? I think all grandmothers are women so no need to add women.

Line 95: WHO/ICO must be written in full at first use.

Line 103: use Ethiopia rather than “our country”

The sentence “Therefore, this study tried to assess the current survival status and predictors of time to death among CC patients in Ethiopia” is confusing. Authors investigated the current survival status and predictors of time to death among CC patients in Ethiopia and did not try to access.

I think authors should structure their introduction to make it interesting and reflect on what they seek to find. At present it is just too difficult to read and understand what they seek to do. This is my suggestion

- Define cervical cancer, its cause, its risk factors and consequences

- Brief epidemiology of cervical cancer worldwide and an emphasis on sub-Saharan Africa and in Ethiopia.

- Trends of cervical cancer deaths in Ethiopia, survival status in Ethiopia.

- What can be done to prevent deaths and what is the situation in Ethiopia

- Review literature on the predictors of time death among women with CC

- Rational for the study and the objectives

Methods

Line 166: remove “which is found”.

Line 121 remove “in” in front of from.

What is the annual population of patients who received care for CC at TASH? If authors are aware of this then their sample size must be reflective of this figure.

Why did authors adopt median survival times in calculating their sample size from a study which reported on clinical trials? This was a cohort study and so authors should explain.

What is “Data were collected by two BSc nurses”? please use standard statements. Data were collected by two registered nurses and not BSc nurses. In fact you can even refer to those who collected data as trained research assistants. What is the motivation to use BSc nurses and MSc nurses? Does it communicate anything?

Regarding the data collection, it is imperative that authors explain succinctly how they were collected. How were the various Socio-demographic Characteristics and Past Obstetrics and medical history collected and their categorization?

Results

Line 163: Should read “continuous variables were reported with mean and standard deviations”.

Line 181: refuse to participate not were refuse to participate.

Line 190 replace whose with who

Discussion

Line 241: authors did not try but they determined or investigated.

First paragraph should briefly summarize the important results of the study. This must be rewritten.

Must be checked for grammatical errors by a native speaker. Statements lie “probability of surviving gets slimed” and many other makes it difficult to read the text.

6. PLOS authors have the option to publish the peer review history of their article (what does this mean?). If published, this will include your full peer review and any attached files.

Reviewer #1: **Yes: **Dan Quansah, PhD.

---

## [Author Response · Author response to Decision Letter 0]

5 Feb 2022

Response to Reviewer 

1. First the wording and grammar should be improved as it is very difficult to read. I suggest a native speaker to edit this manuscript. 

First the wording and grammar should be improved as it is very difficult to read. I suggest a native speaker to edit this manuscript. 

Response: We have tried to correct all the grammatical errors using detailed revision and using online grammar checkers.

Also the introduction is not structured very well and needs to be reworked. 

Response: We have made a revision on Introduction section of the manuscript.

2. Abstract

Introduction:

Authors should provide the full meaning of any abbreviations at the first use. Authors should write out the full meaning of TASH at first use. The phrase in the objective that indicate that authors assess is wrong. Authors investigated or determined. Access is not the right word to use.

Response: The full meaning of abbreviations at the first use is provided. The full meaning of TASH is written at first use as Tikur Anbesa specialized Hospital (TASH)

Response: The phrase in the objective that indicates that assess is corrected by investigated as you commented. 

Methods:

The statement “..sample of 348 patients under follow-up time…” should be rephrased. Authors followed 348 women or a cohort of 348 women. Follow-up time seems vague and not standard. And it is imperative for authors use the phrase “348 patients” rather than a sample of 348 patients. This should be corrected throughout the whole manuscript.

Response: the phrase a sample of 348 patients is corrected by the phrase “348 patients” throughout the whole manuscript as you suggested

 The sentence “The data was collected and entered using the android version CS-Entry tool and for the analysis exported to SPSS version 23” should be broken into two and made clear. Just say data was analyzed by SPSS version 23. Authors should keep their sentences simple and clear.

Response: The sentence is broken into two and made clear. 

The data was collected using the android version CS-Entry tool. Data was analyzed by SPSS version 23.

Conclusions:

The sentences “As the stage of cancer progressed, the chance of surviving gets reduced. Therefore, due emphasis should be given on improving early detection. Routine CC screening programs for high-risk women such as elderly and HIV positive women should be strengthened” should be re written and made more clear and concise. Phrases like “surviving gets reduced” is not scientific and should be written in a better way.

Response: Conclusion is rewritten based on the findings in better way. The death rate of CC patients was high. The significant predictors associated with shorten time to death of CC patients were older age, advanced cancer stage at diagnosis, HIV infection and not receiving cancer treatment. Therefore, improving early detection and initiation of treatment for all CC patients is necessary in order to improve patient’s survival status. The government needs to strengthen the routine CC screening programs to address high-risk women such as elderly and HIV positive women in Ethiopia.

Introduction

Line 65- authors should put “early” in front of screening

The first paragraph of the introduction is succinctly written and the epidemiology is clearly stated. Therefore, I suggest authors delete their second paragraph as it only elaborately gives the epidemiology of Cervical cancer and adds no new information. It makes the introduction unnecessary long and adds no value besides the manuscript is not focused on the epidemiology of CC.

Lines 81-85 already stated in the first paragraph and must not be repeated.

Line 89: what is grandmother women? I think all grandmothers are women so no need to add women.

Line 95: WHO/ICO must be written in full at first use.

Line 103: use Ethiopia rather than “our country”

The sentence “Therefore, this study tried to assess the current survival status and predictors of time to death among CC patients in Ethiopia” is confusing. Authors investigated the current survival status and predictors of time to death among CC patients in Ethiopia and did not try to access.

I think authors should structure their introduction to make it interesting and reflect on what they seek to find. At present it is just too difficult to read and understand what they seek to do. This is my suggestion

- Define cervical cancer, its cause, its risk factors and consequences

- Brief epidemiology of cervical cancer worldwide and an emphasis on sub-Saharan Africa and in Ethiopia.

- Trends of cervical cancer deaths in Ethiopia, survival status in Ethiopia.

- What can be done to prevent deaths and what is the situation in Ethiopia

- Review literature on the predictors of time death among women with CC

- Rational for the study and the objectives

Response: We agreed with your suggestion to reorganize the introduction section and we have made the necessary adjustments. Through corrections made on Introduction section, we have addressed all of your comments and suggestion. We have marked newly added statements for your kind consideration from line 53-103.

Methods

Line 166: remove “which is found”.

Response: “which is found” is removed as suggested

Line 121 remove “in” in front of from

Response: “in” is removed as suggested.

What is the annual population of patients who received care for CC at TASH? If authors are aware of this then their sample size must be reflective of this figure.

Response: We couldn’t get the actual number of the annual population of patients who received care for CC at TASH. But we have tried to show the number of new cases in 2015 ‘In Ethiopia, about 7,095 new CC cases are diagnosed in 2015.’ In the third paragraph of introduction.

Why did authors adopt median survival times in calculating their sample size from a study which reported on clinical trials? This was a cohort study and so authors should explain.

Response: In calculating the sample size median survival times were taken from previous cohort study conducted in black lion not from clinical trials. The error was made during citing the reference.

What is “Data were collected by two BSc nurses”? please use standard statements. Data were collected by two registered nurses and not BSc nurses. In fact you can even refer to those who collected data as trained research assistants. What is the motivation to use BSc nurses and MSc nurses? Does it communicate anything?

Response: we corrected the statement as suggested; Data were collected by two registered nurses. BSc nurses are registered nurses. Data collection process was supervised by senior oncology nurse who has master degree on oncology nursing and who have more experience in supervising research data collection and previously participated in research data collection and supervision.

Regarding the data collection, it is imperative that authors explain succinctly how they were collected. How were the various Socio-demographic Characteristics and Past Obstetrics and medical history collected and their categorization?

Response: the above mentioned information was collected by reviewing the patient card. This information were written in every cancer patient cards based on the standard patient chart prepared by Ethiopian Federal Ministry of Health. 

Results

Line 163: Should read “continuous variables were reported with mean and standard deviations”.

Response: the sentence is corrected as suggested, continuous variables were reported with mean and standard deviations

Line 181: refuse to participate not 

Response: were refuse to participate is corrected with refuse to participate as suggested

Line 190 replace whose with who

Response: whose is replaced with ‘who’ as suggested

Discussion

Line 241: authors did not try but they determined or investigated.

Response: try to assess is corrected with investigated as suggested 

First paragraph should briefly summarize the important results of the study. This must be rewritten. Must be checked for grammatical errors by a native speaker

Response: The important results of the study are briefly summarized in the first paragraph as requested.

Statements lie “probability of surviving gets slimed” and many other makes it difficult to read the text.

Response: We have tried to correct all the grammatical errors using detailed revision and using online grammar checkers.

---

## [Editor Report · Decision Letter 1]

10 Feb 2022

Predictors of time to death among Cervical Cancer patients at Tikur Anbesa Specialized Hospital from 2014 to 2019: A Survival Analysis

PONE-D-21-03938R1

Dear Dr. Seifu,

We’re pleased to inform you that your manuscript has been judged scientifically suitable for publication and will be formally accepted for publication once it meets all outstanding technical requirements.

Kind regards,

James P Brody

Academic Editor

PLOS ONE

---

## [Editor Report · Acceptance letter]

15 Feb 2022

PONE-D-21-03938R1 

Predictors of time to death among cervical cancer patients at Tikur Anbesa specialized hospital from 2014 to 2019: A survival analysis 

Dear Dr. Seifu:

I'm pleased to inform you that your manuscript has been deemed suitable for publication in PLOS ONE. Congratulations! Your manuscript is now with our production department. 

Kind regards, 

on behalf of

Dr. James P Brody 

Academic Editor

PLOS ONE